# Self-Powered Broadband Photodetector Based on NiO/Si Heterojunction Incorporating Graphene Transparent Conducting Layer

**DOI:** 10.3390/nano14060551

**Published:** 2024-03-21

**Authors:** Bhishma Pandit, Bhaskar Parida, Hyeon-Sik Jang, Keun Heo

**Affiliations:** School of Semiconductor and Chemical Engineering, Semiconductor Physics Research Center, Jeonbuk National University, Jeonju 54896, Republic of Korea; bhim2046@jbnu.ac.kr (B.P.); bhaskar.parida@dewa.gov.ae (B.P.)

**Keywords:** graphene/NiO/Si, nanostructure, heterojunction, self-powered, photodetector

## Abstract

In this study, a self-powered broadband photodetector based on graphene/NiO/n-Si was fabricated by the direct spin-coating of nanostructured NiO on the Si substrate. The current–voltage measurement of the NiO/Si heterostructure exhibited rectifying characteristics with enhanced photocurrent under light illumination. Photodetection capability was measured in the range from 300 nm to 800 nm, and a higher photoresponse in the UV region was observed due to the wide bandgap of NiO. The presence of a top graphene transparent conducting electrode further enhanced the responsivity in the whole measured wavelength region from 350 to 800 nm. The photoresponse of the NiO/Si detector at 350 nm was found to increase from 0.0187 to 0.163 A/W at −1 V with the insertion of the graphene top layer. A high photo-to-dark current ratio (≃10^4^) at the zero bias indicates that the device has advantageous application in energy-efficient high-performance broadband photodetectors.

## 1. Introduction

Photodetector devices have become a key component for environmental monitoring, light wave communication, military applications such as missile plum detection, medical treatment, and astronomical observation [1,2,3,4,5,6,7,8]. Instead of single-band detectors, broadband photodetectors such as UV-visible, visible-infrared, and UV-visible-infrared photodetectors are preferred in order to meet the ever-increasing multifunctional requirements. The formation of heterojunction between two materials is advantageous in terms of the suppression of photocarrier recombination and can benefit from different materials’ properties, thus resulting in excellent stability, fast response/recovery time, high responsivity, and enhanced external quantum efficiency [9,10,11,12]. Wide-bandgap metal oxides such as TiO_2_, NiO, WO_3_, Ga_2_O_3_, ZnO, InO, In_2_O_3_, SnO_2_, and their heterostructures with existing semiconductors are also promising for such a wide range of photodetection and sensing application [11,12,13,14,15,16]. The advances in theoretical studies with well-supported experimental results further improve the extended application of metal oxide-based systems [14,15,17]. In addition, high thermal and chemical stability, easy chemical modification, and the availability of both the p-type and n-type lead to the stable operation of metal oxide-based heterojunction in light harvesting, gas sensing, photovoltaics, and photodetection [18,19,20]. Among them, NiO has a wide bandgap (3.4–4.3 eV) and can fulfill such requirements in combination with Si, as reported previously [13,21,22]. Thermal/UV oxidation, e-beam evaporation, RF-magnetron sputtering, pulsed laser deposition, and chemical processes are mainly used to prepare the NiO [23,24,25,26,27]. It was reported that the NiO prepared by a cost-effective sol-gel process has a high performance due to its damage-free construction compared to RF sputtering and others. Parida et al. [27] demonstrated the rectifying characteristics of the nanostructured NiO/p-Si diode showing a photoresponse at wavelengths of 385, 515, and 620 nm. The responsivity of the NiO/Si diode was subsequently enhanced by Zhang et al. using the P (NiO) and n-Si (111) structures, showing the broadband detection from 350 to 600 nm [28]. In another study, Kuru et al. investigated the enhancement of the graphene/Si solar cell using NiO doping [29]. On the other hand, Yang et al. enhanced the power conversion efficiency of the dye-sensitized solar cell using the graphene/NiO composite film and suggested that the loss of photogenerated holes in NiO is due to its low intrinsic electric conductivity and hole diffusion coefficient [30]. Kim et al. reported the high performance of perovskite solar cells by enhancing the electrical conductivity of NiO via copper doping [31]. In this respect, the photoresponse of such a device fabricated using NiO needs a good carrier collecting layer which can enhance the carrier separation and extraction rate. 

A 2-dimensional honeycomb array of a carbon atom, i.e., graphene, has been used as a transparent conducting layer in various optoelectronic and photovoltaic devices, due to its high conductivity, high optical transmittance, and low sheet resistance, thus replacing the expensive transparent conducting metal oxide layer [32,33]. The broadband absorption, ease of functionalization, controlled fabrication process, incorporation of nanomaterials, and formation of heterostructures with existing semiconductors and 2D layers result in the exciting performance of the graphene-based devices [34,35,36,37]. In this study, we have used graphene as a transparent conducting electrode on a NiO/Si heterostructure. The responsivity of the NiO/Si photodiode increased by more than one order by using graphene for all the measured regions at wavelengths of 350–800 nm. The photodiode exhibited a high response at the zero bias, clearly indicating the self-powered performance; furthermore, with increasing bias voltage, the responsivity was increased significantly, which can be attributed to the efficient carrier collection enabled by the graphene top layer. 

## 2. Experiment

The NiO nanoparticles are prepared by a sol-gel process [27]; briefly, Ni(NO_3_)_2_·6H_2_O (0.5 mol) was dispersed in 100 mL of deionized (DI) water, and the PH was adjusted to 10 by adding a NaOH solution. When the obtained green solution had been stirred for 5 min, Ni(OH)_2_ precipitated on the bottom of the flask and was thoroughly washed five times with DI water. The collected green precipitate was dried at 80 °C overnight and ground using a mortar and pestle. Finally, the product was calcinated in a hot tube furnace for 3 h. A prepared nanopowder was mixed in water assisted by sonication for 20 min, and then spin-coated in n-Si [100] with a resistivity of 1–10 Ω cm and annealed for 10 min at 200 °C. The spin-coating process was repeated three times to obtain the desired thickness. Before spin-coating, the Si samples were cleaned with acetone, methanol, IPA, and DI rinse water, each for 5 min. The native SiO_2_ was removed using a buffer oxide etchant, and finally, inorganic particles were removed using a piranha solution (1:1, H_2_SO_4_:H_2_O_2_). Afterward, the graphene was transferred to the NiO/Si sample using a PMMA sacrificial layer. In brief, the PMMA was spin-coated on graphene which was grown on a copper foil and annealed for 5 min at 200 °C. Further, the back-side graphene was removed using 1 M nitric acid solution for 3 min. The copper foil was dissolved completely using ammonium persulfate solution for ~7 h. Further, the PMMA/graphene stack was cleaned with DI water three times, for 20 min each time. Later, the PMMA/graphene stack was transferred to the NiO/Si and allowed to dry naturally; afterward, the sample was annealed for 30 min at 200 °C. Finally, the PMMA was removed by dipping the sample in acetone at 60 °C overnight. By completing the graphene transfer, the back and front contacts were made by an InGa eutectic solution and silver paste, respectively. The schematic diagram of the NiO/Si and graphene/NiO/Si diodes is shown in Figure 1a. Figure 1b shows the Raman spectroscopic measurement of the graphene. Three graphene-related Raman peaks were found around wavenumbers 1580, 2668, and 1340, corresponding to the in-plane vibrational mode G peak, the double resonant 2D peak, and the defect-induced D peak. The calculated intensity ratio of the 2D peaks shows a value higher than unity, implying the monolayer nature of the graphene. The thickness of the NiO was confirmed to be 120 nm by SEM measurements, as shown in Figure 1c.

The current–voltage (I–V) under dark and illuminated conditions was measured using a 2400 Keithley source meter (Tektronix, Seoul, Republic of Korea) in combination with a Bentham SSM150Xe switching monochromator (Bentham, Seoul, Republic of Korea). The transmittance/absorbance of NiO film was measured using a UV/VIS spectrometer (model: JASCO ARSN-917) with the spin-coated NiO on both sides of the polished sapphire substrate. An FESEM image was taken to confirm the nanostructure of the NiO on the sapphire substrate. All electrical characteristics were measured at room temperature and normal ambient conditions.

## 3. Results and Discussion

Figure 2a shows an FESEM image of the NiO nanoparticles spin-coated on both sides of the polished sapphire substrate, illustrating that the NiO nanoparticles were uniformly distributed across the substrate. The size of the NiO nanosheets being lower than 100 nm was confirmed as in the previous study [27]. The 23 nm of RMS roughness of the spin-cast NiO observed from the AFM image indicates that the NiO nanosheet has a mean thickness higher than that of 23 nm, as shown in Figure 2b. The XPS spectra of the NiO before and after graphene transfer, as presented in Figure 2c, shows the two distinct Ni 2p^3/2^ and Ni 2p^1/2^ peaks observed at around 855 and 873.2 eV, respectively; in addition, their corresponding satellite peaks observed at a slightly higher binding energy for the NiO nanoparticles confirms that the NiO quality remains the same after the graphene transfer process [38,39,40]. Figure 2d shows the transmittance spectra of the spin-coated NiO nanoparticles on both sides of the polished sapphire substrate from 300 to 800 nm by considering both sides of the polished sapphire substrate as a reference baseline. The transparency of NiO is much higher in the visible region and shows a gradual increase in transmittance with increasing the wavelength towards the near-infrared region. However, the transparency sharply dropped towards the lower value in the UV region, due to the absorption of light from the higher energy level. Figure 2e shows the absorption spectra of the NiO nanoparticles spin-coated on both sides of the polished sapphire substrate. The absorption increases exponentially when it enters into the UV region (i.e., for lower than 350 nm). This drastic increase in absorption is due to the strong bandgap absorption from the NiO layer, where the energy of the incident photon lies near or higher than the bandgap of NiO [41]. In order to calculate the bandgap energy of NiO, we further plotted the (*αhν*)^2^ versus energy (eV) curve (Tauc plot) and fitted the linear region as shown in Figure 2f. This confirms the corresponding bandgap energy of 3.57 eV that falls on the bandgap region of the NiO (3.4–4.3 eV) reported by various previous studies [13,21,22].

Figure 3 shows the current–voltage (I–V) characteristics measured from the NiO/Si (a) and graphene/NiO/Si (b) diodes in the range of −5 to +5 V under dark conditions and with the illumination of 350 nm. The I–V characteristics curve under dark and illuminated conditions shows clear rectifying properties. Around four orders of the rectification ratio were measured at the forward and reverse bias of 1 V. A low reverse leakage current and a high rectification ratio were observed in comparison to the previous studies on the NiO/Si diode, NiO fabricated by UV oxidation, and sputtering techniques [24,42]. A dark current at 0 V was observed in the range of tens of nano-amperes and gradually increased with increasing the reverse voltage. Further, the increase in the reverse current with the illumination of light at a wavelength 350 nm implies that the UV illumination-induced current was added to the external circuit. The use of graphene on the top of the NiO/Si diode shows a further increase in the total amount of photocurrent in comparison to the device fabricated by using NiO/Si only. For the illumination of light at a wavelength of 350 nm, the photocurrent typically increased from 14 µA to 73 µA in the diode with graphene as the top electrode. The higher conductivity of graphene seems to be much more supportive for the collection of the carrier generated by an incident photon before the electron–hole pair recombination. Figure 3c represents the I–V characteristics of the graphene/NiO/Si photodetector under dark and 300 to 800 nm of illumination with an interval of 50 nm. The enhanced photocurrent under illumination confirms the broadband photodetection characteristics of the fabricated device.

Further, the responsivity of the photodiode was calculated by using the relation *R* = (*I_photo_* − *I_dark_*)/*P_inc_* where *I_photo_* is the photocurrent, *I_dark_* is the dark current, and *P_inc_* is the incident optical power. Here, in our measurement system, the incident optical power was measured as 1.68, 2.31, 3.21, 3.84, 3.61, 3.2, 2.69, 2.1, 1.91, and 1.45 mW/cm^2^ for 350, 400, 450, 500, 550, 600, 650, 700, 750, and 800 nm, respectively. Figure 4a and Figure 4c show the calculated photoresponse values from 350 nm to 800 nm with an interval of 50 nm for the graphene/NiO/Si and NiO/Si photodetectors, respectively. The high responsivity of the graphene/NiO/Si diode in comparison to the NiO/Si diode might be due to the enhanced carrier separation and collection through a fully spanned top graphene electrode [43,44]. The responsivity in terms of wavelength for the graphene/NiO/Si diode shows a decrease in responsivity from higher to lower wavelength regions, which is a similar trend observed by previous groups [27,45]. Different from those previous studies, we observed some elevation in the responsivity curve when it enters the UV region. The variation in the responsivity might be due to the band edge of the materials used. A sharp elevation in responsivity for each measured wavelength was observed for the device containing a graphene top electrode. The amount of responsivity gets reduced when the wavelength reduces, according to the relation $R=\frac{\eta \lambda }{1.24}$ where *R* is the responsivity, *η* is the quantum efficiency (QE), and *λ* is the wavelength in µm. The modified form of the previous equation was used to calculate the efficiency $\eta =\frac{R\;\times \;1.24}{\lambda \;(\mu m)}$ of the photodetector. Therefore, it seems that the plot of QE with respect to λ might be valuable for the comparative study. Figure 4b,d show the QE versus λ plot of the graphene/NiO/Si and NiO/Si photodetectors, respectively, which clearly shows an increase in the bias leading to the higher QE. The higher QE of the NiO/Si photodetector in the UV region implied that the NiO layer mainly contributed to the photocurrent generation. The insertion of the top graphene layer further improved the QE of the device, as shown numerically in Figure 4b; the graphene layer improved the QE from 1.18 to 19.5% at zero bias under the illumination condition of 350 nm. The electron–hole pair produced during the illumination should be collected before the recombination; in this sense, the graphene acted as the carrier transport layer and enhanced the device performance. The highest QE of about 20% at the zero bias condition implies that the efficient self-operating device in the UV region was much higher than previously mentioned. The single detector covering the broad spectral range is promising for a low-cost optical communication system. 

Figure 5a shows the current on–off properties of both the NiO/Si and graphene/NiO/Si diodes at zero volts from 350 to 600 nm with an interval of 50 nm. The photocurrent on–off characteristics observed at 0 V refer to the self-operation of the device without applying any external bias. The self-powered device responding to the broad spectral region has many advantages in the field of energy-efficient optical communication, sensing, and detection. Rapid electron–hole transport and a lower recombination rate were also essential to achieve a high-performance photodetector. Here, by inserting the graphene as a transparent conducting electrode, the value amount of collected photocurrent is increased for all measured wavelengths. The stable switching response to various wavelengths was confirmed under normal ambient conditions from 350 to 600 nm with repetitive on–off cycles. Furthermore, the 50 repetitive on–off cycles were tested under 1 V of reverse bias with the illumination condition of a 500 nm light source, as presented in Figure 5b. Consistent photoresponse without any distortion was observed through all repeated cycles. Figure 5c shows the on–off properties of diodes at the different biasing values of −1, −2, −3, and −5 V, where the increase in photocurrent leads to an increase in the photoresponse value, although the dark current also increased with respect to increasing the reverse bias voltage. Figure 5d presents the current on–off ratio of the photodetector under the illumination of 350 nm and with a bias voltage ranging from 0 to −5 V. The current on–off ratio increases as the applied reverse bias voltage is reduced. The highest on–off ratio of ~10^5^% was observed at 0 V, indicating the high switching ratio under self-powered conditions. The self-powered operation of the NiO/Si photodetector can be understood clearly from the band diagram presented in Figure 6. Once the p-type NiO comes in contact with n-Si, the band bending occurs to meet the equilibrium position and fermi level aligned by the diffusion of the majority of carriers to the lower concentration regions. The band bending at the interface leads to the formation of a depletion region and a built-in electric field directed from the Si to the NiO. Upon the illumination of light having an energy higher than the Si or NiO bandgap energy, the electron and holes are generated and separated by the field present at the interface. The photogenerated holes are moved toward the valence band of the NiO, and the electrons are moved toward the conduction band of the Si and contribute to the enhanced current in the external circuit. Under reverse bias conditions, the external electric field is aligned with the built-in electric field and significantly increases the depletion region, hence enhancing the carrier generation and increasing the photocurrent.

Detectivity, one of the most important merits of photodetection, signifies the minimum signal of a photon that can be detected by the photodetector and is given by the following relation [46,47]: $D=\frac{A^{\frac{1}{2\;}}R}{{\left(2qI_{D}\right)}^{\frac{1}{2}}}$ where *A* is the active area of PD, *q* is the unit electronic charge, and *I_D_* is the dark current. Under 350 nm of illumination, the detectivity of 1.66 × 10^12^, 2.71 × 10^11^, 2.23 × 10^11^, 1.98 × 10^11^, and 1.78 × 10^11^ cmHz^1/2^W^−1^ was observed at biases of 0, −1, −2, −3, and −5 V, respectively. The decrease in the detectivity of the photodetector with increasing the reverse bias voltage corresponds to the effective increase in the dark current in comparison to the photocurrent. Maximum detectivity of the photodetector at 0 V signifies the efficient operation under self-powered conditions. The conductivity of NiO nanoparticles by doping, or by optimizing the device structure for the performance of the graphene/NiO/Si diode, can be improved further [47,48,49,50,51,52]. In our study, we discuss fruitful information about the graphene conducting layer’s effects on the NiO/Si diode, which provides essential information for future investigation. The photodetection characteristics and parameters of the NiO/Si photodetectors along with the previous studies are summarized in Table 1 [53,54,55,56,57,58,59,60,61].

## 4. Conclusions

In conclusion, we successfully employed graphene as a transparent electrode in the NiO/Si broadband photodetector by using the CVD graphene, NiO nanoparticles, and n-type Si. Both the NiO/Si and graphene/NiO/Si diodes show rectifying properties and photoresponse towards the broad spectral region. I–V characteristics were measured with different illumination conditions ranging from 350 to 800 nm, representing the enhanced performance of the photodetector. The high conductivity of the graphene electrode enhances the carrier collection efficiency before the recombination of induced carriers and enhances the QE by up to 78%. The ~20% of QE at 0 V bias for a wavelength of 350 nm illustrates the efficient self-powered performance of the graphene/NiO/Si heterojunction photodetector.

## Figures and Tables

**Figure 1 nanomaterials-14-00551-f001:**
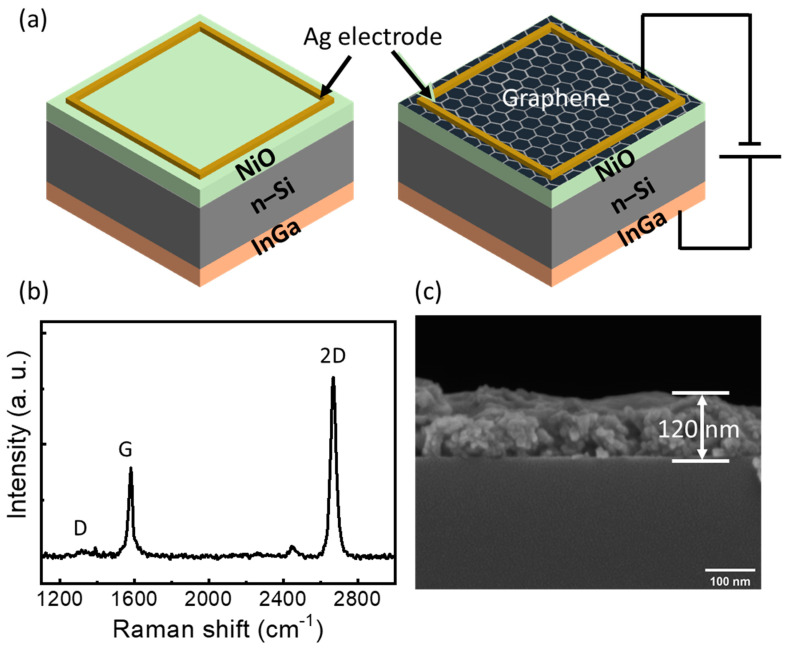
(**a**) Schematic diagram of the fabricated NiO/n-Si diode without (**left**) and with the graphene layer (**right**); (**b**) Raman spectroscopic measurement of graphene; (**c**) SEM image of NiO on the Si substrate for thickness measurement.

**Figure 2 nanomaterials-14-00551-f002:**
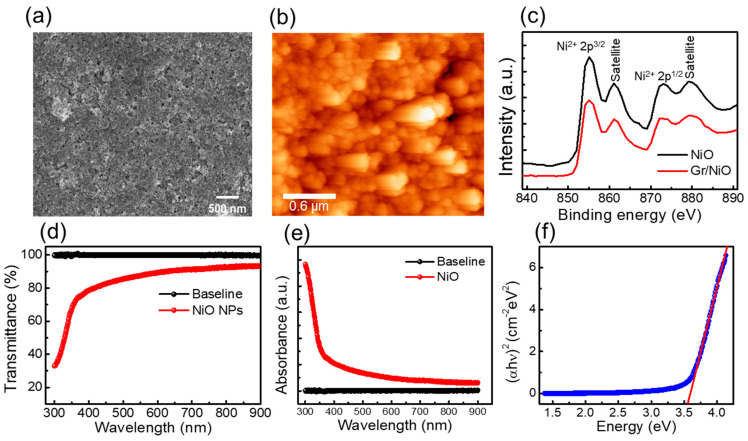
(**a**) FESEM image of the NiO nanoparticles coated on the Si surface; (**b**) AFM image of the NiO nanoparticles; (**c**) XPS analysis of the NiO nanoparticles before and after the graphene transfer; (**d**) transmittance spectra and (**e**) absorption spectra of the NiO coated on both sides of the polished sapphire substrate as a baseline; and (**f**) Tauc plot of the NiO nanoparticles with linear fitting (red line).

**Figure 3 nanomaterials-14-00551-f003:**
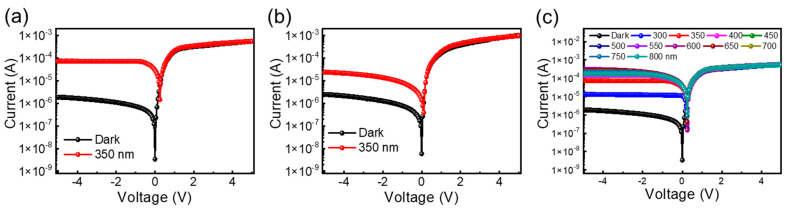
Current–voltage (I–V) characteristics of the (**a**) graphene/NiO/Si and (**b**) NiO/Si diodes under dark and illuminated (350 nm) conditions; (**c**) I–V characteristics of the graphene/NiO/Si photodetector under various illumination wavelengths.

**Figure 4 nanomaterials-14-00551-f004:**
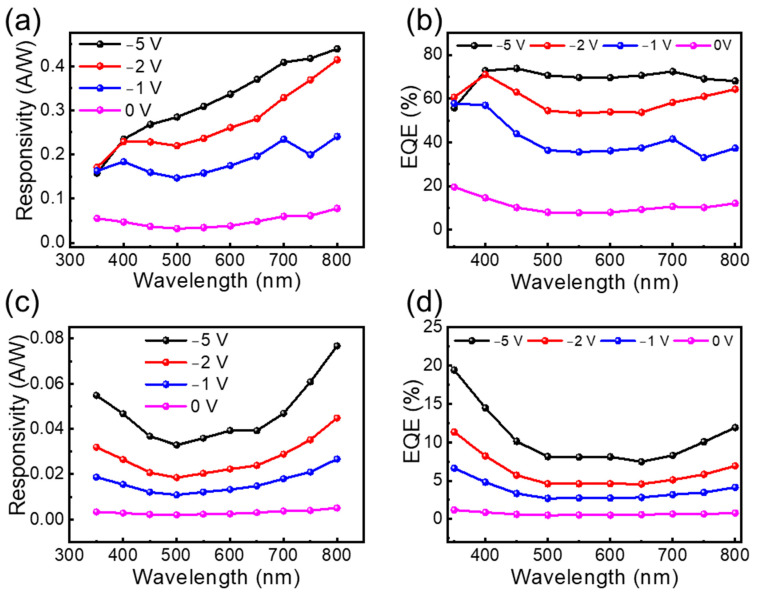
The responsivity (**a**) and corresponding QE (**b**) of the graphene/NiO/Si diode, and the responsivity (**c**) and QE (**d**) of the NiO/Si diode, at various reverse bias conditions.

**Figure 5 nanomaterials-14-00551-f005:**
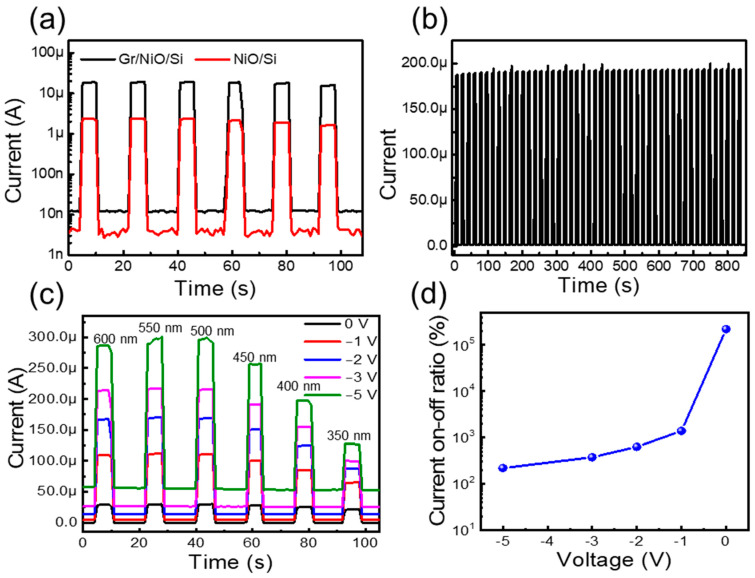
Temporal current on–off characteristics measurement of the (**a**) NiO/Si and graphene/NiO/Si diodes under a zero bias condition; (**b**) the 50 times-repeated cycle of the graphene/NiO/Si diode at −2 V bias under 500 nm illumination; (**c**) the on–off response with various reverse bias conditions from 350 to 600 nm; (**d**) the current on–off ratio under the illumination of 350 nm light at various bias conditions.

**Figure 6 nanomaterials-14-00551-f006:**
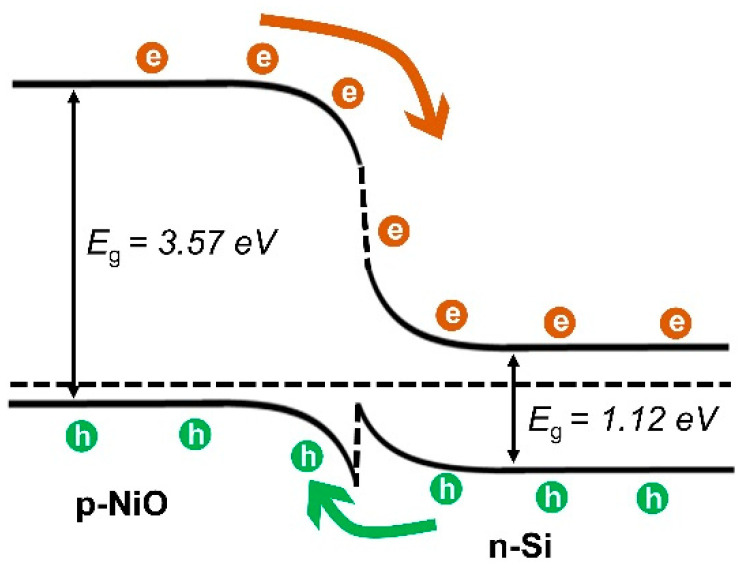
The energy band diagram of the NiO/Si heterojunction.

**Table 1 nanomaterials-14-00551-t001:** Comparison of the NiO/Si photodetectors with previous studies.

PD	Responsivity (A/W)	Dark Current (A)	Wavelength (nm)	Photocurrent	T_rise_ (s)	T_fall_ (s)	Detectivity (cmHz^1/2^W^−1^)	Bias (V)	Ref.
Al/NiO/Si/Al	0.7	NA	330	NA	NA	NA	63 × 10^12^	−3	[53]
NiO/n-Si	~0.15@0 V ~0.17@30 V	NA	290	NA	NA	NA	NA	0	[45]
Ni/NiO/p-Si/InGa	0.156 1 V	3.9 × 10^−8^	385	7.1 × 10^−7^	0.5	NA	NA	−3	[27]
Au/NiO/Si MSM	~6.5	2.32 μA	365	NA	0.11	NA	NA	5	[54]
Au/NiO/Si MSM	4.5	~5 μA	365	NA	226 ms	200 ms	NA	5	[55]
Ag/Zn:NiO/p-Si/Ag	0.249	~13 μA cm^−2^ (−1 V)	350	79 μA cm^−2^	0.3	NA	2.3 × 10^11^ (−1 V)	−4	[56]
Ag/NiO/n-Si/Ag	0.43	7 μA cm^−2^ (−0.2 V)	600	2.5 × 10^5^	30 ms	NA	1.5 × 10^10^ (0 V)	−2	[28]
NiO/n-Si	0.013	3.9 × 10^−9^	365	NA	85 ms	85 ms	1.03 × 10^11^	0	[57]
Al/NiO/Si MSM	91 μA/mW	59 μA	350	NA	750 ms	800 ms	NA	0.2	[58]
NiO/n-Si PD	0.83	9.1 × 10^−12^	365	2.8 × 10^−8^	0.1	0.1	NA	0	[59]
NiO NW/n-Si(100)	9.1 mA W^−1^	NA	350	NA	0.4	NA	1.8 × 10^9^	0	[60]
NiO/n-Si	160 mA W^−1^	NA	365	NA	1.5	NA	NA	5	[61]
graphene/NiO/n-Si	0.055	3.35 × 10^−9^	350	2.9 × 10^−5^	<0.8	<0.8	1.66 × 10^12^	0	This work
	0.15	1.82 × 10^−6^		7.3 × 10^−5^			1.78 × 10^11^	−5	

## Data Availability

Data are contained within the article.

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
