# Peer review of "Self-Powered Broadband Photodetector Based on NiO/Si Heterojunction Incorporating Graphene Transparent Conducting Layer"

_nanomaterials, 2024, doi:10.3390/nano14060551_

Round 1

Reviewer 1 Report

Comments and Suggestions for Authors

In this paper, the authors prepared a self powered broadband photodetector based on graphene/NiO/n-Si by directly spin coating nanostructured NiO on a Si substrate. The photodetection capability has been measured from the 300 nm to 800 nm range, and higher photo-response in the UV region was observed due to the wide bandgap of NiO. I believe that publication of the manuscript may be considered only after the following issues have been resolved.

1.    What is the thickness of graphene in this work? I don't think the authors have provided any corresponding characterization.

2.    In this device, nickel oxide is not grown in situ. What is the adhesion strength of the relevant thin films? This is crucial for the stability of the device.

3.    Suggest the author to provide relevant structural parameters in Figure 1.

4.    In order to increase the readability of the article, in the introduction section, the author needs to mention some relevant latest references on the application of graphene-based material, such as, Diamond and Related Materials 142, 2024, 110793; Opto-Electron Sci 2, 230012 (2023); Opto-Electron Adv 5, 200098 (2022).

5.    The English expression of the whole article needs to be further improved.

Comments on the Quality of English Language

Minor editing of English language required

Reviewer 2 Report

Comments and Suggestions for Authors

This manuscript suggests a self-powdered broadband photodetector employing a NiO/Si heterojunction. Importantly, a graphene transparent conducting layer is also employed in the configuration. Main feature is the declared photodetection capability from the 300 nm to 800 nm range, with a higher photo-response in the UV region observed, due to the wide bandgap of NiO. Also notable is that the configuration exhibits high photo-to-dark current ratio (≃104) at the zero bias. The choice of configuration, and the measurement approaches and characteristics revealed are valuable and attracting broad research interest. However, existing studies in the field that chart relevant theoretical/modeling guidance at different levels of theory and/or structural/morphological knowledge for similar material configurations exist but are not yet referenced in the present manuscript.

Good quality of figures and concise discussion are other positive features of this manuscript

The present manuscript raises some minor technical concerns and some linguistic concerns too. There are questions related to insufficient introduction of theoretical relevance. These points amount to a minor to moderate revision before acceptance of this manuscript for publication:

1: Title is too long; the authors should consider shortening it. It is incorrect in English to begin each word in a title with a capital letter. In addition, the word “consisting” is wrongly used both in meaning, the authors want to say “incorporating”, and grammatically… all in all the title needs to be written anew and formatted to be short.

2: The abstract should clearly and explicitly mention the measurements techniques of relevance.

3: Throughout the text, the authors call the graphene layer “graphene transparent conducting layer”, the sequence of adjectives and the choice of terminology are not optimal. In their configuration, this is a capping layer, it should be called “a capping layer of graphene”, and only once in the text, in its beginning, it may be clarified that the layer is both conductive and transparent.

4: Thermal stability and thermal properties are important aspect of the metal oxide nanomaterials, they should be clearly and explicitly discussed in the context of the present heterojunction and also in the context of photodetector applications.

5: The authors fail to mention the recent literature dedicated to studies of metal oxide heterostructures/heterojunctions by models employing DFT and other levels of theory, works in which it is shown how theory guides the experimental realization and verification of the properties of this class of material systems, e.g., CrystEngComm 23 (2021) 6661-6667; and Applied Surface Science 548 (2021) 149275. Such concepts and existing literature should be reflected in the introduction.

Comments on the Quality of English Language

The manuscript still needs a comprehensive grammatical revision.

Round 2

Reviewer 1 Report

Comments and Suggestions for Authors

Accept in present form